# Variations in Growth and Photosynthetic Traits of Polyploid Poplar Hybrids and Clones in Northeast China

**DOI:** 10.3390/genes13112161

**Published:** 2022-11-19

**Authors:** Luping Jiang, Xiangzhu Xu, Qun Cai, Rui Han, Mulualem Tigabu, Tingbo Jiang, Xiyang Zhao

**Affiliations:** 1State Key Laboratory of Tree Genetics and Breeding, School of Forestry, Northeast Forestry University, Harbin 150040, China; 2College of Forestry and Grassland Science, Jilin Agricultural University, Changchun 130118, China; 3College of Veterinary Medicine, Jilin University, Changchun 130062, China; 4Tree Seedling Management Station, Forestry Department of Jilin Province, Changchun 130607, China; 5Southern Swedish Forest Research Center, Swedish University of Agricultural Science, P.O. Box 49, 230 52 Lomma, Sweden

**Keywords:** polyploid poplar, hybrid clones, selective breeding, comprehensive evaluation

## Abstract

To evaluate differences among 19 different ploidy hybrid poplar clones grown in northeast China, 21 traits related to growth traits and photosynthetic characteristics were detected and analyzed. Abundant phenotypic variations exist among and within populations, and these variations are the basis of forest tree genetic improvements. In this research, variance analysis showed that the traits except the net photosynthesis rate among the different ploidies and all the other traits exhibited significant differences among the ploidies or clones (*p* < 0.01). Estimation of phenotypic coefficients of variation, genotypic coefficients of variation, and repeatability is important for selecting superior materials. The larger the value, the greater the potential for material selection improvement. The repeatability of the different traits ranged from 0.88 to 0.99. The phenotypic and genotypic coefficients of variation of all the investigated traits ranged from 6.88% to 57.40% and from 4.85% to 42.89%, respectively. Correlation analysis showed that there were significant positive correlations between tree height, diameter, and volume. Transpiration rate, intercellular carbon dioxide concentration, and stomatal conductance were significantly positively correlated with each other but negatively correlated with instantaneous water use efficiency. Growth traits were weakly correlated with photosynthetic indexes. The rank correlation coefficient showed that most of the growth indicators reached a significant correlation level among different years (0.40–0.98), except 1-year-old tree height with 4-year-old tree height and 1-year-old ground diameter with 3-year-old tree height, which indicated the potential possibility for early selection of elite clones. Principal analysis results showed that the contribution rate of the first principal component was 46.606%, and 2-year-old tree height, 2-year-old ground diameter, 3-year-old tree height, 3-year-old ground diameter, 3-year-old diameter at breast height, 3-year-old volume, 4-year-old tree height, 4-year-old ground diameter, 4-year-old diameter at breast height, and 4-year-old volume showed higher vector values than other traits. With the method of multiple-trait comprehensive evaluation to evaluate clones, SX3.1, SY3.1, and XY4.2 were selected as elite clones, and the genetic gains of height, basal diameter, diameter at breast height, and volume of selected clones ranged from 12.85% to 64.87% in the fourth growth year. The results showed fundamental information for selecting superior poplar clones, which might provide new materials for the regeneration and improvement of forests in Northeast China.

## 1. Introduction

Poplars (*Populus* spp.) are one of the most important economic tree species due to their rapid growth [1], short rotation [2], adaptability [3], and excellent wood properties [4] in temperate regions of the world [5]. Genetic improvement research on poplar has been conducted for nearly 80 years [6], and many improved varieties have been selected and used widely [7], playing an important role in pulpwood [8], plywood [9], medium-density fiberboard [10], water conservation effects [11], and road greening [12].

Polyploidization is considered a crucial force in plant species evolution and speciation, which may occur during the processes of plant growth and development and play important roles in plant diversity [13,14,15]. Generally, the increase in chromosome ploidy levels contributes to the increase in gene dose and cell volumes [16,17], which promote polyploid advantages in biochemical, morphological, and physiological processes in trees, ultimately leading to larger growth traits [14]. Polyploid breeding has great application potential in the breeding of new forest varieties with fast growth, high quality, and high stress resistance. Many research results have indicated that triploid poplar has a higher photosynthetic index and growth traits than diploid and tetraploid trees [18]. Wang et al. [19] found that there was a correlation between photosynthetic traits and wood property traits. Tang et al. [20] found that growth traits were positively correlated with wood traits, and the indirect evaluation of polyploid materials through photosynthetic and growth traits could provide a guarantee for the improvement of wood yield and quality. In particular, the triploid *Populus tomentosa* cultivated at Beijing Forestry University grows rapidly in North China; the volume of 8 years is 2–3 times that of diploid control [21], which contributes a great contribution to local economic development [22].

In Northeast China, high seasonal precipitation (70–80% occurring from June to September) and low average annual temperatures (4.0–5.5 °C) are salient climatic features that are key factors limiting poplar growth and productivity. In addition, the lack of superior poplar varieties and clones is also the primary constraint restricting poplar cultivation in Northeast China [23]. At present, only a small number of varieties (*Populus simonii* × *Populus nigra* ‘Xiaohei’, *Populus nigra* × *Populus simonii* ‘Zhonglin Sanbei-1’, etc.) were selected in the 1980s and have been widely used for afforestation in the past 40 years [24]. With the high-frequency asexual propagation and frequent occurrence of extreme weather in recent years, the varieties have shown a serious decline in trend, which greatly limits the ecological and economic benefits to Northeast China. Therefore, the selection of new adaptable varieties with rapid growth and good wood quality is significant for poplar breeding.

In this study, the growth traits and photosynthetic characteristics of 19 poplar clones of different ploidies were investigated. Several genetic and variation parameters were calculated, and elite clones were ultimately selected based on principal analysis. This research might provide new materials for regeneration in Northeast China.

## 2. Materials and Methods

### 2.1. Experimental Site

The experiment was performed at the Yushutai forest farm (43°15′ N, 124°41′ E; altitude 171 m), which is located in Lishu County, Siping city, Jilin Province. The climate was a northern temperate semihumid continental monsoon climate. The precipitation ranged from 420 to 480 mm, and the annual average temperature was 3.4 °C in poplar and the ecological environment in arid areas. The contents of carbon, nitrogen, phosphorus, and potassium in soil were 16.62 mg/g, 1.65 mg/g, 0.66 mg/g and 18.63 mg/g, respectively.

### 2.2. Experimental Materials

Several main diploid poplar cultivars (Table 1) were used as parents, and artificially controlled pollination was performed in 2012. The external morphology of the flower bud, disk, and another was observed by stereopicroscope, and the fixed pollen mother cells were observed under an optical microscope to determine the relationship between the development process of pollen mother cells and the external morphology. Furthermore, flower buds were selected during the diakinesis stage in meiosis, and then treated with 0.5% colchicine 4 times with 2 h intervals to obtain polyploid pollen. The crossing experiment was carried out with reduplicated pollen to obtain polar seeds with different ploidies. After sowing, the DNA content in the leaves was evaluated by flow cytometry using the method of Zhao et al. [18]. After three years of seedling growth, the hybrid progeny was preliminarily selected through seedling height and ground diameter, and 16 polyploid poplar hybrid clones with tree height and ground diameter over 20% of the average were selected as super seedlings for further experiments. Among them, 13 clones were triploid (HB3.1, XY3.1, XY3.4, XY3.5, XY3.6, XY3.7, SX3.1, SX3.2, SX3.3, SY3.1, SY3.2, HY3.1, HY3.3), and 3 clones were tetraploid (HB4.1, XY4.1, XY4.2). To evaluate the utilization potential of the selected clones, three diploid poplar clones (XH, XH14, BCXH) mainly planted in local areas were selected as the control for the experiment. The experimental forest was established using one-year-old trees in the spring of 2017. The experimental design was a completely randomized design with thirty duplicate trees at a spacing of 2.0 × 4.0 m.

### 2.3. Measurements of Traits and Statistical Analyses

The tree height (H, m) and basal diameter (BD, mm) of all living intact trees were measured using box staff and digital calipers after falling leaves from 2017 to 2020, and the diameter at breast height (DBH, mm) was measured in 2019 and 2020. The stem volume (*V*, m^3^) was calculated based on DBH (*D*_1.3_) and H using equation [25]:$$V=0.000041960698\times D_{1.3}^{1.9094595}\times H^{1.0413892}$$

Six trees were selected from each clone, and the 5th to 10th leaves from the upper lateral branches of the different trees were picked off in August 2017. Leaf length (LL, mm) and leaf width (LW, mm) were measured by vernier caliper, and the leaf area (LA, cm^2^) of flattened leaves were measured using ImageJ software (https://imagej.nih.gov/ij/index.html accessed on 16 November 2022). The leaf shape index (*LSI*) was calculated according to the formula:LSI=leaf length/leaf width

Six trees from each clone were selected to determine the photosynthetic parameters. The net photosynthesis rate (*P_n_*, μmol·m^−2^·s^−1^), intercellular CO_2_ concentration (*C_i_*, μmol·mol^−1^), stomatal conductance (*G_s_*, mol·m^−2^·s^−1^) and transpiration rate (*T_r_*, mol·m^−2^·s^−1^) of the 5th to 7th leaves of the south upper lateral branch were investigated using a portable photosynthesis system (Li-6400, LI-COR, Lincoln, NE, USA) from 8:30 am to 11:30 am on sunny days in August 2017. The photosynthetic photon flux density and ambient CO_2_ concentration were 1400 μmol∙m^−2^ and 400 μmol∙mol^−1^, respectively. The water use efficiency (*WUE*) was calculated according to the formula:$$WUE=P_{n}/T_{r}$$

Statistical analysis was performed using SPSS 25.0, R (https://www.r-project.org/ accessed on 16 November 2022), and DPS software (https://www.theaccessgroup.com/en-gb/our-brands/dps-software/ accessed on 16 November 2022). The soil conditions of the planted forestland were consistent, and a variance analysis was performed using the single-plant data without considering the block effect. The significance of the fixed effects was tested by analysis of variance (ANOVA) *F* tests. Variations were analyzed using equation [26]:$$Y_{ijk}=µ+\alpha_{i}+\beta_{j\left(i\right)}+\varepsilon_{ijk}$$
where µ is the overall mean, $\alpha_{i}$ is the effect of ploidy type, $\beta_{j\left(i\right)}$ is the effect value of the *j* clone in the *i* ploidy type, and $\varepsilon_{ijk}$ is the random error.

The phenotypic (*PCV*s, %) and genotypic (*GCV*s, %) coefficients of variation were estimated as follows [27]:$$PCV=\frac{\sqrt{\sigma_{p}^{2}}}{\overset{¯}{X}}\times 100\%$$
$$GCV=\frac{\sqrt{\sigma_{g}^{2}}}{\overset{¯}{X}}\times 100\%$$
where $\overset{¯}{X}$ is the mean average of the trait, $\sigma_{P}^{2}$ is the total phenotypic variance between individuals, and $\sigma_{g}^{2}$ is the genetic variance component between individuals.

The repeatability (*R*) of the traits was calculated as follows [28]:$$R=\frac{\sigma_{A}^{2}}{\sigma_{A}^{2}+\sigma_{b}^{2}+\sigma_{e}^{2}}$$
where $\sigma_{A}^{2}$ is the genotypic component, $\sigma_{b}^{2}$ is the clone variance and $\sigma_{e}^{2}$ is the error variance component.

The phenotypic correlation coefficient ($r_{A\;}\left(xy\right))$ of traits x and y was calculated according to [29]:$$r_{A\;}\left(xy\right)=\frac{COV_{P\left(x,y\right)}}{\sigma_{p\left(x\right)}\;\sigma_{p\left(y\right)}}$$
where $COV_{p\left(x,y\right)}$ is the covariance between traits x and y, $\sigma_{p\left(x\right)}$ is the variance component for trait x, and $\sigma_{p\left(y\right)}$ is the variance component for trait y.

The Spearman correlation coefficient ($r_{s}$) among traits in different years was calculated according to [30]:$$r_{S}=\frac{6\sum_{i=1}^{N}d_{i}^{2}}{N\left(N^{2}-1\right)}$$
where N is the number of clones and $d_{i}^{2}$ is the square of rank.

The principal component value and comprehensive score were computed using the following formulas [31]:$$Y_{i}=\overset{n}{\underset{j=1}{\sum }}\alpha_{ij}X_{j}\left(j=1,2,3,\cdots n\right)$$
$$W=\overset{p}{\underset{i=1}{\sum }}Y_{i}\omega_{i}\left(i=1,2,3,\cdots ,p\right)$$
where $Y_{i}$ is the i-th principal component value, $\alpha_{ij}$ is the eigenvalue of trait j in principal component i, $X_{j}$ is the average value of trait  j, W is the comprehensive score, $\omega_{i}\;$ is the contribution rate of the i-th principal component, n is the number of traits, and p is the number of extracted principal components.

Genetic gain (ΔG) was estimated using the formula [32]:$$\Delta G=R\Delta W/\bar{X}\times 100\%$$
where R and $\bar{X}$ are the repeatability and mean values of the traits, respectively. ΔW is the selection difference.

## 3. Results

### 3.1. Genetic and Variation Parameters

Variance analysis showed that there were significant differences among ploidies or clones (*p* < 0.01), except for *P_n_* among ploidy (*p* >0.05) based on *F* tests (Table 2). The phenotypic coefficients of variation (*PCV*s) of all traits ranged from 6.88% to 57.40% and were higher than the genotypic coefficients of variation (*GCV*s), which ranged from 4.85% to 42.89%. Volume should have the highest *PCV* or *GCV* values [33,34], which were higher than 40%, but the traits *C_i_*, *P_n_*, LL, LW, and *LSI* showed lower *PCV* and *GCV* values, which were lower than 10%. The repeatability (*R*) of the different traits ranged from 0.88 to 0.99 (Table 3) and showed high values, which indicated that the traits were stable within clones among different ramets. The *PCV* and *GCV* values of H and BD showed an upward trend, followed by a downward trend with tree growth.

### 3.2. Comparison of Each Trait’s Average Values

Different ploidy effects are shown in Appendix A. For leaf traits or photosynthesis traits, the average LL, LW, LA, *P_n_*, and *WUE* of triploid clones were higher than the average values of the control or tetraploid clones, but for growth traits, such as H, BD, and *V*, in different growth years, control clones showed higher values than triploid or tetraploid clones.

The average LL and LA of all poplar clones were 127.01 mm and 134.29 cm^2^, respectively. The clone SY3.1 exhibited the highest values (148.44 mm, 171.60 cm^2^), which were higher than the lowest clone XY3.1 (108.87 mm, 91.40 cm^2^) by 36.34% and 87.75%, respectively. The average value of LW was 147.80 mm, and the highest clone SY3.2 (181.79 mm) was higher than the lowest clone XY3.1 (121.41), at 49.73%. The average value of the *LSI* was 0.86, and the *LSI* of clone SX3.1 (1.02) was 41.67% higher than that of clone SY3.2 (0.72) (Appendix A). Among the different ploidies, the average values of LL and LA in the triploids were higher than those in the diploids and tetraploids.

In terms of photosynthetic characteristics, the average values of *P_n_*, *C_i_*, *G_s_*, *T_r_*, and *WUE* were 19.24 μmol·m^−2^·s^−1^, 277.96 μmol·mol^−1^, 0.40 mol·m^−2^·s^−1^, 4.49 mol·m^−2^·s^−1^, and 4.43, respectively. The highest clones were SX3.3 (22.22 μmol·m^−2^·s^−1^), SX3.2 (301.89 μmol·mol^−1^), SX3.2 (0.55 mol·m^−2^·s^−1^), SX3.3 (5.72 mol·m^−2^·s^−1^), and SX3.1 (5.45), whereas the lowest clones were SY3.2 (16.17 μmol·m^−2^·s^−1^), HB4.1 (253.67 μmol·mol^−1^), XY3.7 (0.29 mol·m^−2^·s^−1^), XY3.7 (3.43 mol·m^−2^·s^−1^), and BXCH (3.60). The highest values were higher than the lowest values at 37.42%, 19.01%, 89.66%, 67.76%, and 51.39%, respectively (Appendix A).

The comparison of growth traits showed that the highest H and BD average values of the first-year-old seedlings were 46.90% and 43.28% higher than the lowest average values, respectively. Clone SX3.1 exhibited the largest H and BD values from the second to the fourth year. The highest H (BD) values of the second-, third-, and fourth-year-old seedlings were 69.47% (81.25%), 65.95% (92.27%), and 55.77% (99.50%), which were higher than the lowest values. The highest DBH (*V*) values of the third- and fourth-year-old seedlings were 120.26% (530.83%) and 112.00% (418.20%), respectively, which were higher than the lowest average values (Appendix A).

### 3.3. Intertrait Correlation Analysis

The relationships among the different traits were determined using Pearson’s correlation analysis. For the 21 traits, we divided them into three types (growth types, leaf types, and photosynthetic types). In Figure 1, a total of 109 significant correlation coefficients were detected, of which 61 combinations were significantly correlated among the growth types. Among leaf types, 5 combinations were significantly correlated, and among the types of photosynthesis, 9 combinations were significantly related. All growth traits reached a highly significant positive correlation level in the same year (0.380–0.977). Among the different years, except for the weak correlation between the annual seedlings and other years, the other years had extremely significant positive correlation levels. Significant negative correlations were found between *LSI* and LW or LA (−0.665–−0.324), whereas significant positive correlations were found among LL, LW, and LA (0.578–0.862). There were significant positive correlations among *G_s_*, *C_i_*, and *T_r_* (0.681–0.861) but negative correlations among *WUE* and *G_s_*, *C_i_*, and *T_r_* (−0.906–−0.616). Meanwhile, LL, LW, and LA were significantly positively correlated with H1, BD1, H2, and BD2, and *WUE* was significantly positively correlated with *LSI*, H4, and *V*4. The *LSI* reached a highly significant correlation level with H3, H4, DBH4, and *V*4.

Spearman’s rank correlation coefficients were further calculated. The correlation coefficients of growth traits at different ages ranged from 0.34 (H1 with H4) to 0.98 (DBH4 with *V*4). Specifically, the correlation coefficients of H at different ages ranged from 0.34 to 0.81, and the correlation coefficients of BD at different ages ranged from 0.61 to 0.92 (Figure 2).

### 3.4. Principal Component Analysis

Principal component analysis (PCA) was performed after data transformation, and three principal components with high eigenvalues were obtained, with a cumulative contribution rate of 81.30% (Appendix A). The contribution rate of the first principal component (PC I) was 46.61%, where growth traits H2, BD2, H3, BD3, DBH3, V3, H4, BD4, DBH4, and *V*4 showed high and positive values. The contribution rate of the second principal component (PC II) was 18.27%, where the positive characteristic vector values of LW and LA were higher than those of the other traits. The contribution rate of the third principal component (PC III) was 16.42%, where the positive characteristic vector values of *G_s_*, *C_i_*, and *T_r_* were higher than those of the other traits.

### 3.5. Elite Clone Selection and Genetic Gain

All traits of 19 clones were detected for comprehensive evaluation using the comprehensive scores and individual component scores of each principal component. In the first principal component scores, SX3.1, SY3.1, and XY4.2 exhibited higher values (Figure 3). The genetic gains of the selected clone were 12.42% (H2), 17.76% (BD2), 10.87% (H3), 16.30% (BD3), 20.29% (DBH3), 51.62% (*V*3), 12.85% (H4), 23.42% (BD4), 24.33% (DBH4), and 64.87% (*V*4) for each trait (Table 4). In the second principal component scores, HY3.3, SY3.1, and SY3.2 displayed higher values (Figure 3). The genetic gains of the selected clone were 17.46% (LW) and 24.70% (LA) (Table 4). In the third component scores, SX3.2, SY3.1, and BCXH showed higher values (Figure 3). The genetic gains of the selected clone were 22.19% (*G_s_*), 5.65% (*C_i_*), and 7.83% (*T_r_*) for each trait.

## 4. Discussion

### 4.1. Genetic and Variation Parameters

Analysis of variance is one of the most effective methods for estimating the degree of variation in breeding populations [35], which helps characterize phenotypic variations and variance components among individual plants [36]. In our study, variance analysis showed that there were significant differences among ploidies or clones, except for *P_n_* among ploidy. The results suggested that abundant variations in these traits were found among ploidy or clones, indicating the great potential of these clones for genetic improvement [37]. Variation parameters, such as phenotypic coefficients of variation (*PCV*s), genotypic coefficients of variation (*GCV*s), and repeatability, are basic elements for breeding research [4,38], and large genetic variation and high repeatability are important for selecting superior materials in specific areas [28]. In this study, the *PCV*s and *GCV*s of the growth traits of all clones ranged from 9.98% to 57.40%, which represents a moderate or high variance coefficient [33,34]. The *PCV* values of *V* (56.70–57.40) showed the largest values, which were higher than those reported for *Populus ussuriensis* [25], indicating that the superior traits evaluated and identified using these methods represent a significant improvement in tree volume.

The trends in the coefficients of variation and repeatability with increasing age were important for determining the appropriate age during early selection, and they could also estimate the selection effects [39]. The *PCV*s and *GCV*s of tree height and basal diameter first increased, then decreased, and finally changed gradually with tree growth. The values of H were lower than BD for different ages, suggesting that trait BD is more effective than H in the early selection of superior clones [40]. The success of tree selection in breeding programs depends on the sufficiently high heritability of the traits, which is a significant indicator in evaluating trait stability [41]. In this research, the repeatability of growth traits ranged from 0.88 to 0.96, indicating that growth traits were highly genetically controlled [33] and that the superior clones selected might exhibit fairly higher genetic gain. In our study, *R* values of H and BD showed an increasing-decreasing-increasing trend with tree growth. The results were the same as those of *Populus* [42] and *Cunninghamia lanceolata* [43], which may be due to the interaction between genetic and environmental factors. Leaf traits are closely related to plant growth strategies, which can accurately reflect the survival strategies of plant adaptation to severe environmental changes [44]. Leaf area determines the size of the photosynthetic area of plants [45], while the leaf aspect ratio contributes to the ability of plants to adapt to ambient temperature and humidity [46]. In the present study, the *R* values of LA and *LSI* were higher, and the *PCV* and *GCV* values of *LSI* were lower than those of LA, which might be due to the genetic composition of the hybrid offspring derived from *P. nigra* and *P. simonii*.

### 4.2. Mean Values

As a widespread biological process, polyploidization has provided much genetic variation for plant adaptive evolution. It not only provides extra gene copies, strengthening the robustness against malignant mutations but also provides abundant genetic materials for neofunctionalization. Therefore, polyploidy has been considered an important force in the evolution of plants [47]. Triploid trees often show higher growth potential and longer leaves than their diploid counterparts. For example, the leaf indices of triploid and tetraploid *P. ussuriensis* are higher than those of diploids, which show a rising trend with increasing ploidy [48]. In this study, triploid clones showed higher averages of LL, LA, *P_n_*, and *WUE*, which was the same result as in other research. This may indicate that plants with an increased number of chromosomes may cause some morphological changes by influencing various physiological activities and metabolic processes [49]. However, the Pn and *WUE* values of the triploid were higher than those of the diploid and tetraploid; these results were the same as those of research on hybrid poplar clones [26], but in *P. ussuriensis*, triploid is the largest, diploid is the smallest, and tetraploid is the center [48]. The variations might reflect differences in leaf structure, composition, transcriptional level, methylation of DNA, or nucleolus dominance in clones of different ploidies, which should be explored in future research [50]. The research also showed that triploids showed lower average H, BD, and *V* values in different growth years than control clones, which was different from other research [18]. The main reason was that the triploid clones were selected in different cross combinations, and not all triploid clones were excellent. Parents also existed, which led to different phenotypes among the different cross combinations. The results indicated that elite clone selection should be based on pedigree selection results [51]. However, most triploid clones had high growth on the basis of mean analysis, among which SX3.1 was significantly better than the diploid clones, demonstrating that SX3.1 exhibited high selection values, thereby being suitable for afforestation. The clones were ranked on the basis of growth traits, and the order was similar after being planted from two years to four years, which was distinguished from one-year-old seedlings. This might be determined by the slow seedling phenomenon of one-year-old plants [52]. The results suggested that there was a certain limitation in the field experiment using one-year-old seedlings as materials compared with seedlings over two years old.

### 4.3. Correlation

Tree species growth and development are complex processes in which various organs participate to control the plant phenotype [53]. The correlation coefficient can represent the correlation level between different traits, which plays an important role in understanding the relationship between different detected traits [54,55]. Thus, the exploration of trait correlations has great significance in forest growth. Age is also a considerable factor in early selection [56]. In our study, the results of the Pearson correlation coefficient in different years showed that most growth traits reached an extremely significant correlation level, which was coincident with the results for *Radiata pine* [57] and *Populus deltoides* [58]. These studies indicated that the early growth of plants had a continuous impact on later development. Pn was positively correlated with *T_r_* and *G_s_*, but the correlation did not reach a significant level, probably because photosynthetic changes in plants are a complex process in which internal and external factors act together [59,60]. The photosynthetic changes in the materials in this study may be mainly influenced by environmental factors. Significant positive correlations were found between LL, *LSI*, *WUE*, and some growth traits in the later stage, while negative correlations were found between *T_r_* and H3, *V*3, DBH4, H4, and *V*4. These results showed that the mentioned traits could be considered advantageous phenotypic traits in poplar multiple-trait selective breeding programs, and Tr could be used as a negative auxiliary index for early material evaluation. The comprehensive evaluation of different traits is beneficial for the selection of superior poplar clones [61].

Knowledge of the correlation between tree height, ground diameter, diameter at breast height, and volume in different years after afforestation is decisive for trait selection and the age of early selection [43]. In this study, the correlation coefficient between different traits in different years showed that most growth traits reached extremely significant correlation levels, which indicated the potential feasibility of early selection of superior clones. The rank correlation between the same traits in different years (H1–H4, BD1–BD4) showed that strong rank correlations between the same years were found, and the rank correlations decreased with increasing time intervals. The results were similar to *Pinus massoniana* [62] and *Larix olgensis* [63]. The rank correlation coefficient between ground diameter, diameter at breast height, and volume showed that the diameter at breast height was more important for early evaluation. In addition, rank correlations were found between H1, BD1, and *V*4, indicating the important role of one-year growth quantity. In total, these results suggested that the appropriate early selection age for poplar was two or three years old, while it was based on adopting the strategy of two-stage evaluation and selection after two years. The ground diameter was considered the basis for primary selection in the first stage, and DBH was used as the standard in the second stage or in the final selection. This method ensured the reliability of selection, which was conducive to superior poplar clones undergoing productive afforestation in advance.

### 4.4. Principal Component Analysis

Principal component analysis is a common multivariate analysis method that helps to reduce the data dimension and preserve data trends and patterns [64]. As a comprehensive evaluation index of clones, the principal component value can accurately express the comprehensive performance of various traits, thereby being widely used in forest evaluation and selection [65,66]. In this study, three principal components reflected 89.379% of the original data, which was similar to the study of *P. ussuriensis* [67] and *Aigeiros* clones [68]. In our study, different principal components were representative of different categories of investigated traits, such as PCI represented growth traits, PCII represented leaf traits, and PCIII represented photosynthetic traits. A similar result was found in the research of Wang et al. [29], who found that PCI represented growth traits, PCII represented wood traits, and PCIII represented trunk form traits in *Piuns koraiensis* clones. Therefore, it was very significant and feasible to use different PC values to comprehensively evaluate the growth traits.

### 4.5. Comprehensive Assessment and Genetic Gain

Breeding targets determine the research methods used in tree genetics [69]. The results of the correlation analysis in this study showed that there was a certain correlation between early leaf and photosynthetic traits and growth traits. Therefore, a comprehensive evaluation of clones by comprehensive growth, photosynthesis, and leaf indicators is of great significance for accurate evaluation of clones. According to the results of the principal component analysis, the overall evaluation of clones was carried out by using the character load value of each principal component as the coefficient and the information content of the original data as the weight. Based on the comprehensive scores, three clones (SX3.1, SY3.2, and XY4.2) were selected as elite clones, with a selection rate of 15%. Genetic gain is a key standard for evaluating the breeding effect due to the excessive extent of the breeding population compared with the existing population [70]. In this study, compared with the other clones, three excellent clones had higher genetic gains in growth indicators, leaf indicators, and photosynthetic indicators. This indicated that the elite clones had good growth performance, which could provide superior material for timber production.

## 5. Conclusions

In conclusion, the growth traits, leaf traits, and photosynthetic characteristics of polyploid poplar hybrid clones were analyzed. Significant differences exist among the ploidy of poplar clones. High *PCV*, *GCV*, and *R* values indicated that elite clone selection was stable and meaningful. The correlation coefficient showed that different traits exhibited significant or weak correlations, which indicated that tree growth was the result of the coordination and restriction of different traits. Principal component analysis showed different principal components instead of different traits, which could be used for elite clone selection. Finally, SX3.1, SY3.1, and XY4.2 were selected as elite clones based on growth traits according to principal component I. These results showed an effective method for selecting superior poplar clones, and the candidate clones might provide new materials for the regeneration and improvement of forests in northeast China.

## Figures and Tables

**Figure 1 genes-13-02161-f001:**
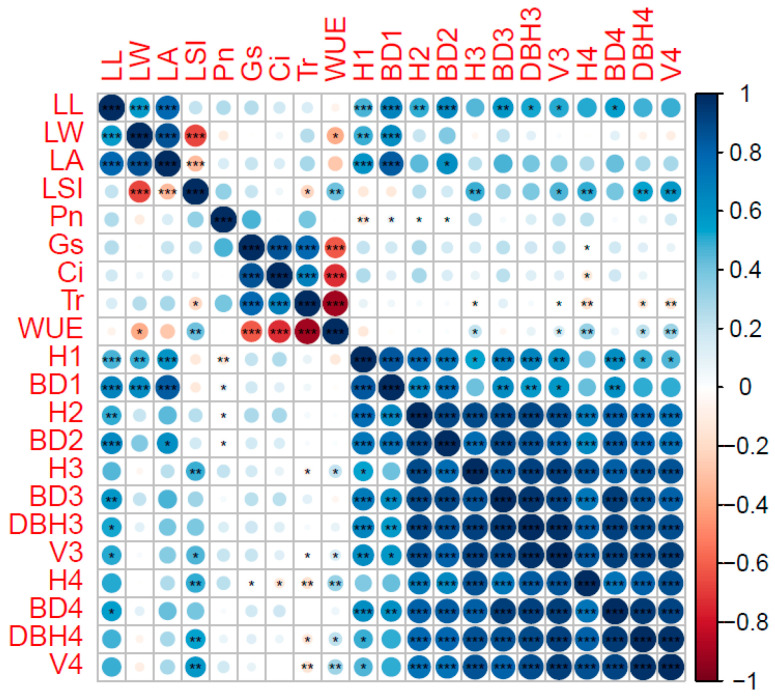
Pearson correlation coefficients among different traits. Note: (*P_n_*): net photosynthesis rate; (*C_i_*): intercellular CO_2_ concentration; (*G_s_*): stomatal con-ductance; (*T_r_*): transpiration rate; (*WUE*): water use efficiency; (LL): leaf length; (LW): leaf weight; (LA): leaf area; (*LSI*): leaf shape index; (H1): 1-year-old tree height; (H2): 2-year-old tree height; (H3): 3-year-old tree height; (H4): 4-year-old tree height; (BD1): 1-year-old ground diameter; (BD2): 2-year-old ground diameter; (BD3): 3-year-old ground diameter; (BD4): 4-year-old ground diameter; (DBH3): 3-year-old diameter at breast height; (DBH4): 4-year-old diameter at breast height; (*V*3): 3-year-old volume; (*V*4): 4-year-old volume; *: Significant correlation at the 0.05 level (1-tailed); **: Significant correlation at the 0.01 level (2-tailed); ***: Significant correlation at the 0.001 level (2-tailed).

**Figure 2 genes-13-02161-f002:**
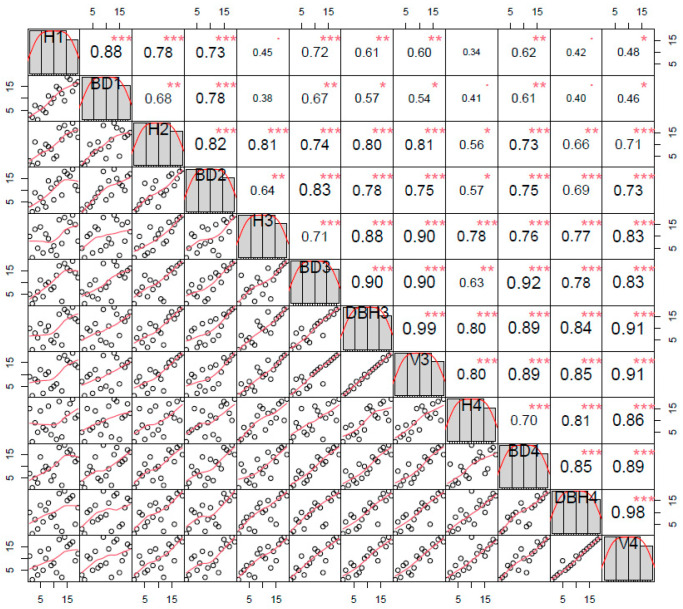
Rank correlation coefficients among different traits. Note: (H1): 1-year-old tree height; (H2): 2-year-old tree height; (H3): 3-year-old tree height; (H4): 4-year-old tree height; (BD1): 1-year-old ground diameter; (BD2): 2-year-old ground diameter; (BD3): 3-year-old ground diameter; (BD4): 4-year-old ground diameter; (DBH3): 3-year-old diameter at breast height; (DBH4): 4-year-old diameter at breast height; (*V*3): 3-year-old volume; (*V*4): 4-year-old volume; *: Significant correlation at the 0.05 level (1-tailed); **: Significant correlation at the 0.01 level (2-tailed); ***: Significant correlation at the 0.001 level (2-tailed).

**Figure 3 genes-13-02161-f003:**
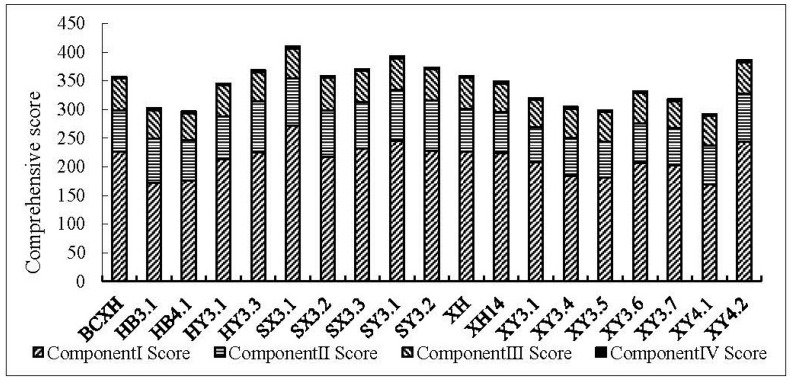
Comprehensive scores of different clones.

**Table 1 genes-13-02161-t001:** Number and name of poplar clones.

Source	Cross Combination	Code
Multiploid	(*P. nigra* × *P. simonii*) × (*P. simonii* × *nigra* ‘*B6*’)	HB3.1, HB4.1
	(*P. simonii* × *P. nigra*) × (*P. nigra* × *P. simonii* ‘Zhonglin Sanbei-1’)	XY3.1, XY3.4, XY3.5, XY3.6, XY3.7, XY4.1, XY4.2
	(*P. deltoides* × *P. nigra* cv., ‘*shandis 1*’) × (*P. simonii* × *P. nigra*)	SX3.1, SX3.2, SX3.3
	(*P. deltoides* × *P. nigra* cv., ‘*shandis 1*’) × (*P. nigra* × *P. simonii* “Zhonglin Sanbei-1”)	SY3.1, SY3.2
	(*P. nigra* × *P. simonii*) × (*P. nigra* × *P. simonii* ‘Zhonglin Sanbei-1’)	HY3.1, HY3.3
Control	*P. simonii* × *P. nigra* ‘*Xiaohei*’	XH
	*P. simonii* × *P. nigra* cv.-14	XH14
	*P. simonii* × *P. nigra* ‘*Baicheng 1*’	BCXH

**Table 2 genes-13-02161-t002:** Variation analysis of growth traits.

Tree Ages	Traits	Variation Source	*SS*	*df*	*MS*	*F*	*Sig*
1	*P_n_*	Ploidies	8.942	2	4.471	2.673	0.072
		Clones	242.703	17	15.169	9.069	0.000
	*C_i_*	Ploidies	2603.146	2	1301.573	6.850	0.001
		Clones	29,202.456	17	1825.153	9.606	0.000
	*G_s_*	Ploidies	0.075	2	0.038	7.670	0.001
		Clones	0.845	17	0.053	10.734	0.000
	*T_r_*	Ploidies	2.682	2	1.341	5.794	0.004
		Clones	82.634	17	5.165	22.313	0.000
	*WUE*	Ploidies	4.001	2	2.001	7.647	0.001
		Clones	74.671	17	4.667	17.840	0.000
	LL	Ploidies	196.612	2	98.306	8.722	0.000
		Clones	9537.561	17	596.098	52.888	0.000
	LW	Ploidies	1381.965	2	690.983	52.699	0.000
		Clones	22,243.955	17	1390.247	106.031	0.000
	LA	Ploidies	2094.263	2	1047.132	50.369	0.000
		Clones	52,480.706	17	3280.044	157.777	0.000
	*LSI*	Ploidies	0.062	2	0.031	45.204	0.000
		Clones	0.448	17	0.028	40.827	0.000
	H1	Ploidies	1.134	2	0.567	19.432	0.000
		Clones	8.551	17	0.503	17.233	0.000
	BD1	Ploidies	40.849	2	20.425	5.609	0.004
		Clones	768.524	17	45.207	12.415	0.000
2	H2	Ploidies	11.561	2	5.780	51.216	0.000
		Clones	49.619	17	2.919	25.861	0.000
	BD2	Ploidies	1188.035	2	594.017	20.235	0.000
		Clones	11,578.139	17	681.067	23.200	0.000
3	H3	Ploidies	10.208	2	5.104	17.961	0.000
		Clones	53.331	17	3.137	11.040	0.000
	BD3	Ploidies	3447.691	2	1723.845	12.493	0.000
		clones	19,963.911	17	1174.348	8.511	0.000
	DBH3	Ploidies	1520.343	2	760.171	12.494	0.000
		Clones	10,930.117	17	642.948	10.567	0.000
	*V*3	Ploidies	0.000	2	0.000	11.397	0.000
		Clones	0.001	17	0.000	10.387	0.000
4	H4	Ploidies	10.998	2	5.499	14.636	0.000
		Clones	95.784	17	5.634	14.996	0.000
	BD4	Ploidies	3632.942	2	1816.471	12.494	0.000
		Clones	39,929.279	17	2348.781	16.155	0.000
	DBH4	Ploidies	1067.266	2	533.633	5.536	0.005
		Clones	21,225.208	17	1248.542	12.952	0.000
	*V*4	Ploidies	0.000	2	0.000	4.829	0.009
		Clones	0.006	17	0.000	13.033	0.000

Note: (*SS*): Sum of Squares of Deviations; (*MS*): mean square; (*df*): degree of freedom; (*F*): *F* value in *F*-test; (*Sig*): Significant level; (*P_n_*): net photosynthesis rate; (*C_i_*): intercellular CO_2_ concentration; (*G_s_*): stomatal conductance; (*T_r_*): transpiration rate; (*WUE*): water use efficiency; (LL): leaf length; (LW): leaf weight; (LA): leaf area; (*LSI*): leaf shape index; (H1): 1-year-old tree height; (H2): 2-year-old tree height; (H3): 3-year-old tree height; (H4): 4-year-old tree height; (BD1): 1-year-old ground diameter; (BD2): 2-year-old ground diameter; (BD3): 3-year-old ground diameter; (BD4): 4-year-old ground diameter; (DBH3): 3-year-old diameter at breast height; (DBH4): 4-year-old diameter at breast height; (*V*3): 3-year-old volume; (*V*4): 4-year-old volume.

**Table 3 genes-13-02161-t003:** Genetic and variation parameters of different traits.

Tree Ages	Traits	Average	SD	*GCV*	*PCV*	*R*
1	*P_n_*	19.24	1.73	6.36	9.26	0.89
	*C_i_*	277.96	18.89	4.85	6.94	0.90
	*G_s_*	0.40	0.10	18.15	25.18	0.91
	*T_r_*	4.49	0.84	16.51	19.68	0.96
	*WUE*	4.43	0.83	15.80	19.58	0.94
	LL	127.01	10.61	6.35	6.88	0.98
	LW	147.80	16.18	8.37	8.72	0.99
	LA	134.29	24.44	14.17	14.57	0.99
	*LSI*	0.86	0.08	6.37	7.05	0.98
	H1	1.98	0.27	11.60	14.46	0.94
	BD1	16.75	2.63	12.83	17.16	0.92
2	H2	3.83	0.61	14.58	17.02	0.96
	BD2	44.43	9.16	19.15	22.70	0.96
3	H3	5.64	0.79	9.98	13.75	0.91
	BD3	73.30	16.10	14.64	21.71	0.88
	DBH3	46.65	11.26	17.24	24.01	0.91
	*V*3	0.005	0.003	41.01	57.40	0.90
4	H4	7.37	0.98	10.37	13.29	0.93
	BD4	95.61	19.51	16.36	20.66	0.94
	DBH4	66.06	14.69	17.13	22.68	0.92
	*V*4	0.013	0.007	42.89	56.70	0.92

Note: (SD): Standard Deviation; (*GCV*): genotypic coefficients of variation; (*PCV*): phenotypic coefficients of variation; (*R*): repeatability; (*P_n_*): net photosynthesis rate; (*C_i_*): intercellular CO_2_ concentration; (*G_s_*): stomatal conductance; (*T_r_*): transpiration rate; (*WUE*): water use efficiency; (LL): leaf length; (LW): leaf weight; (LA): leaf area; (*LSI*): leaf shape index; (H1): 1-year-old tree height; (H2): 2-year-old tree height; (H3): 3-year-old tree height; (H4): 4-year-old tree height; (BD1): 1-year-old ground diameter; (BD2): 2-year-old ground diameter; (BD3): 3-year-old ground diameter; (BD4): 4-year-old ground diameter; (DBH3): 3-year-old diameter at breast height; (DBH4): 4-year-old diameter at breast height; (*V*3): 3-year-old volume; (*V*4): 4-year-old volume.

**Table 4 genes-13-02161-t004:** Comprehensive evaluation of elite clones.

Elite Clone	Traits	Mean Values of Elite Clones	Mean Values of Control Clones	Mean Values of Population	Mean Values of Elite Clones Higher Than Control (%)	Genetic Gain (%)
HY3.3, SY3.1, and SY3.2	LW	153.28	139.00	147.8	25.08	17.46
	LA	160.97	123.67	134.29	35.58	24.70
SX3.2, SY3.1, and BCXH	*G_s_*	0.42	0.44	0.40	13.19	22.19
	*C_i_*	280	286.70	277.96	3.06	5.65
	*T_r_*	4.31	4.74	4.49	2.42	7.83
SX3.1, SY3.1, and XY4.2	H2	4.32	4.07	3.83	6.32	12.42
	BD2	52.68	46.65	44.43	12.92	17.76
	H3	6.31	5.90	5.64	7.03	10.87
	BD3	86.83	81.07	73.3	7.11	16.30
	DBH3	57.11	51.70	46.65	10.46	20.29
	*V*3	0.008	0.007	0.005	26.19	51.62
	H4	8.39	7.68	7.37	9.16	12.85
	BD4	119.48	104.67	95.61	14.15	23.42
	DBH4	83.48	69.89	66.06	19.45	24.33
	*V*4	0.0230	0.015	0.013	51.02	64.87

Note: (LW): leaf weight; (LA): leaf area; (*G_s_*): stomatal conductance; (*C_i_*): intercellular CO_2_ concentration; (*T_r_*): transpiration rate; (H2): 2-year-old tree height; (H3): 3-year-old tree height; (H4): 4-year-old tree height; (BD2): 2-year-old ground diameter; (BD3): 3-year-old ground diameter; (BD4): 4-year-old ground diameter; (DBH3): 3-year-old diameter at breast height; (DBH4): 4-year-old diameter at breast height; (*V*3): 3-year-old volume; (*V*4): 4-year-old volume.

## Data Availability

Data are contained within the article and Appendix A.

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
