# Peer review of "Variations in Growth and Photosynthetic Traits of Polyploid Poplar Hybrids and Clones in Northeast China"

_genes, 2022, doi:10.3390/genes13112161_

Round 1
Reviewer 1 Report
This study focuses in the improvement of the economic poplar taxa and deals with an important experimental work as breeding studies concerning trees is time consuming and more complex as for others agronomic species. The manuscript was well written, the objectives are clear, the methodology is appropriate for this study. However some clarifications and improvements are needed for this manuscript.
-The authors are invited to not use abbreviations before their explanations (like Pn in the abstract, ts for temperature in experimental section).
-Correct stomatalconductance to stomatal conductance.
Line 25: correct “but negatively positively correlated with instantaneous???
Line 46: correct to “is considered as”.
-Correct to: the increase in chromosomes ploidy level.
- Correct this sentence “These factors led to slow poplar growth of 16.62 76 mg/g, 1.65 mg/g, 0.66 mg/g and 18.63 mg/g, respectively.” Respectively to what ??
-The point which required more clarification is the used plant material: the taxonomic status of the used plant material should be clarified in the title, in the abstract and experimental section. Does the parent poplar cultivars used in breeding are triploid??
-In the title you can add “hybrids and clones”
-Please provide more information about the induction of polyploidy pollens.
-Some minor errors in English language should be corrected to improve the manuscript which is well written.
Author Response
Response to Reviewer 1 Comments
Thank you for your comments. These comments are all valuable and very helpful for revising and improving our paper, as well as the important guiding significance to our researches. We have studied comments carefully and have made correction which we hope meet with approval. The reply is as follows.
Point 1: The authors are invited to not use abbreviations before their explanations (like Pn in the abstract, ts for temperature in experimental section).
Response 1: Thanks for your comments. It is really true as your suggested. We are very sorry for our negligence. We have made correction according to your comments.
Please see line 21, and line 35-38 in the revised manuscript.
Special thanks to you for your good comments.
Point 2: Correct stomatalconductance to stomatal conductance.
Response 2: Thanks for your comments. It is really true as your suggested. We are very sorry for our negligence. We have made correction according to your comments.
Please see line 29 in the revised manuscript.
Special thanks to you for your good comments.
Point 3: Line 25: correct “but negatively positively correlated with instantaneous???
Response 3: Thanks for your comments. It is really true as your suggested. We are very sorry for our negligence. We have made correction according to your comments.
The changes to the manuscript are as follows:
“Transpiration rate, intercellular carbon dioxide concentration and stomatal conductance were significantly positively correlated with each other, but negatively correlated with instantaneous water use efficiency.”
Please see line 29 in the revised manuscript.
Special thanks to you for your good comments.
Point 4: Line 46: correct to “is considered as”.
Response 4: Thanks for your comments. It is really true as your suggested. We are very sorry for our negligence. We have made correction according to your comments.
Please see line 107 in the revised manuscript.
Special thanks to you for your good comments.
Point 5: Correct to: the increase in chromosomes ploidy level.
Response 5: Thanks for your comments. It is really true as your suggested. We are very sorry for our negligence. We have made correction according to your comments.
Please see line 109-110 in the revised manuscript.
Special thanks to you for your good comments.
Point 6: Correct this sentence “These factors led to slow poplar growth of 16.62 76 mg/g, 1.65 mg/g, 0.66 mg/g and 18.63 mg/g, respectively.” Respectively to what??
Response 6: Thanks for your comments. It is really true as your suggested. We are very sorry for our negligence. We have made correction according to your comments.
The changes to the manuscript are as follows:
“The contents of carbon, nitrogen, phosphorus, and potassium in soil were 16.62 mg/g, 1.65 mg/g, 0.66 mg/g and 18.63 mg/g, respectively”.
Please see line 144-145 in the revised manuscript.
Special thanks to you for your good comments.
Point 7: The point which required more clarification is the used plant material: the taxonomic status of the used plant material should be clarified in the title, in the abstract and experimental section. Does the parent poplar cultivars used in breeding are triploid??
Response 7: Thanks for your comments. It is really true as your suggested. The breeding parent poplar variety is diploid. We are very sorry for our negligence. We have made correction according to your comments.
The changes to the manuscript are as follows:
“Several main diploid poplar cultivars (Table 1) were used as parents, and artificially con-trolled pollination was performed in 2012”.
Please see line 147 in the revised manuscript.
Special thanks to you for your good comments.
Point 8: In the title you can add “hybrids and clones”
Response 8: Thanks for your comments. It is really true as your suggested. We have made correction according to your comments.
The changes to the title are as follows:
“Variations in growth and photosynthetic traits of polyploid poplar hybrids and clones in Northeast China”.
Please see line 3 in the revised manuscript.
Special thanks to you for your good comments.
Point 9: Please provide more information about the induction of polyploidy pollens.
Response 9: Thanks for your comments. It is really true as your suggested. We have made correction according to your comments.
The changes to the manuscript are as follows:
“The external morphology of flower bud, disk and anther was observed by stereopicroscope, and the fixed pollen mother cells were observed under the optical microscope to determine the relationship between the development process of pollen mother cells and the external morphology. Furthermore, flower buds were selected during diakinesis stage in meiosis, and then treated with 0.5 % colchicine 4 times with 2 hours interval, to obtain polyploid pollen”.
Special thanks to you for your good comments.
Point 10: Some minor errors in English language should be corrected to improve the manuscript which is well written.
Response 10: Thanks for your comments. It is really true as your suggested. We apologize for the poor language of our manuscript. We have now worked on both language and readability, and we invited a professional teacher to edit the language and proofread the grammar of our manuscript again. We really hope that the flow and language level have been substantially improved. Many revisions have been presented in the revised manuscript.
Special thanks to you for your good comments.
Reviewer 2 Report
The study evaluated variations in 19 variant hybrid poplar clones based on 21 traits related to growth and photosynthesis.
Generally, the paper is well written, scientifically sound and relevant. some few suggestion and comments to improve the manuscript:
Authors should mention the growth and parameters in the abstract, before the results highlights, for ease of understanding to the readers.
Alot of assumption in the abstract, the reader is not privy of what was done e.g what is Pn?
Line 24,use a full stop after volume, rather than a semicolon, otherwise, the sentence is misleading.
Line 25...but negatively positively correlated....is confusing, authors school revise.
Semi colon is misused. it is confusing and makes the sentences too lengthy.
Line 28....significant correlation level of? quote the value.
line 28 to 29, which year was the significant correlation level attained?
the authors should indicate which traits are represented by H2, BD2, H3, BD3....otherwise the abstract is not easy to understand.
Line 34-36, to my understanding, the authors results provide fundamental information that would guide selection of polar clone it does not provide the best method but rather , information.
Line 55-59 is too long, authors should fragment . sentence at productivity . shortages of ....
In the introduction, the authors fail to highlight the importance of poplar which makes validates the study. Why is poplar important to human or the ecology.
Which of the studied parameters contribute to wood quality? please add this is the introduction.
Line 67, include a full stop after......investigated . Then start several.......as a new sentence.
The authors need to review on the benefits conferred to trees through polyploidy.
Line 76 the values of poplar growth are they with respect to years? or clones?
Line 86, what was the basis of selecting the 16 polyploidy poplar hybrid trees?
Line 93, check the correct name of the experimental design used by the authors.
In the experimental design,. did the authors factor in local control?
Description of the field experiment is vague. Did the authors evaluate the utilization potential of the 13 clones ? With the three diploid poplar clones as the controls? if so, why the six duplicate trees in the experiment?
The height and basal diameter and breast height were measured at what intervals?
The photosynthetic parameters were measured on sunny days: for how long?
The variance analysis did not consider the blocking effect, was this an assumption that blocking had no effect ? if so, why didn't the authors use CRD instead?
In their statistical model (line 121) the authors are not clear why they have effect due to clones (??) , at the same time, effect of the clone (??(?)). authors need to clarify.
In line 163,, authors write that volume shared have the highest PCV or GCV values, reference should be provided.
The footnote under table 2 should not stand for two tables.no matter the repetitiveness, every table and figure should stand alone.
Table 4 , The authors divided the traits into three broad categories, instead of 'different traits' mention the three traits.
Table 5 title is incomplete . Tables should stand alone . authors should mention what the genetic gains are for
Discussion: 4.2 mean values , the authors results on Pn and WUE contradicted those on populus ussuriensis. Authors should briefly state the p. urruriensis findings.
Line 370 sounds incomplete, consider revising it
Line 386-389, interpret the study findings with the information in line 384-386
More comments and suggestions are in the manuscript attached.

Author Response
Response to Reviewer 2 Comments
Thank you for your comments. These comments are all valuable and very helpful for revising and improving our paper, as well as the important guiding significance to our researches. We have studied comments carefully and have made correction which we hope meet with approval. The reply is as follows.
Point 1: Authors should mention the growth and parameters in the abstract, before the results highlights, for ease of understanding to the readers.
Response 1: Thanks for your comments. It is really true as your suggested. We are very sorry for our negligence. We have made correction according to your comments.
The changes to the manuscript are as follows:
“Abundant phenotypic variation exists among and within populations, and these variations are the basis of forest tree genetic improvements. In this research, variance analysis showed that the traits except net photosynthesis rate among the different ploidies and all the other traits exhibited significant differences among the ploidies or clones (P < 0.01). Estimation of phenotypic coefficients of variation, genotypic coefficients of variation and repeatability is important for selecting superior materials. And the larger the value, the greater the potential for material selection improvement. The repeatability of different traits ranged from 0.88 to 0.99. The phenotypic and genotypic coefficients of variation of all the investigated traits ranged from 6.88% to 57.40% and 4.85% to 42.89%, respectively”.
Please see line 19-24 in the revised manuscript.
Special thanks to you for your good comments.
Point 2: A lot of assumption in the abstract, the reader is not privy of what was done e.g what is Pn?
Response 2: Thanks for your comments. It is really true as your suggested. We are very sorry for our negligence. We have made correction according to your comments.
Please see line 21, and line 35-38 in the revised manuscript.
Special thanks to you for your good comments.
Point 3: Line 24,use a full stop after volume, rather than a semicolon, otherwise, the sentence is misleading.
Response 3: Thanks for your comments. It is really true as your suggested. We are very sorry for our negligence. We have made correction according to your comments.
Please see line 28 in the revised manuscript.
Special thanks to you for your good comments.
Point 4: Line 25...but negatively positively correlated....is confusing, authors school revise.
Response 4: Thanks for your comments. It is really true as your suggested. We are very sorry for our negligence. We have made correction according to your comments.
The changes to the manuscript are as follows:
“Transpiration rate, intercellular carbon dioxide concentration and stomatal conductance were significantly positively correlated with each other, but negatively correlated with instantaneous water use efficiency.”
Please see line 29 in the revised manuscript.
Special thanks to you for your good comments.
Point 5: Semi colon is misused. it is confusing and makes the sentences too lengthy.
Response 5: Thanks for your comments. It is really true as your suggested. We are very sorry for our negligence. We have made correction according to your comments.
Please see line 30 in the revised manuscript.
Special thanks to you for your good comments.
Point 6: Line 28....significant correlation level of? quote the value.
Response 6: Thanks for your comments. It is really true as your suggested. We are very sorry for our negligence. We have made correction according to your comments.
Please see line 32 in the revised manuscript.
Special thanks to you for your good comments.
Point 7: line 28 to 29, which year was the significant correlation level attained?
Response 7: Thanks for your comments. It is really true as your suggested. We are very sorry for our negligence. We have made correction according to your comments.
The changes to the manuscript are as follows:
“The rank correlation coefficient showed that most of the growth indicators reached significant correlation level among different years (0.40-0.98), except 1-year-old tree height with 4-year-old tree height and 1-year-old ground diameter with 3-year-old tree height, which indicated the potential possibility for early selection of elite clones”.
Please see line 31-33 in the revised manuscript.
Special thanks to you for your good comments.
Point 8: the authors should indicate which traits are represented by H2, BD2, H3, BD3....otherwise the abstract is not easy to understand.
Response 8: Thanks for your comments. It is really true as your suggested. We are very sorry for our negligence. We have made correction according to your comments.
Please see line 35-38 in the revised manuscript.
Special thanks to you for your good comments.
Point 9: Line 34-36, to my understanding, the authors results provide fundamental information that would guide selection of polar clone it does not provide the best method but rather , information.
Response 9: Thanks for your comments. It is really true as your suggested. We are very sorry for our negligence. We have made correction according to your comments.
The changes to the manuscript are as follows:
“The results showed a fundamental information to select superior poplar clones, which might provide new materials for the regeneration and improvement of forests in Northeast China”.
Please see line 42 in the revised manuscript.
Special thanks to you for your good comments.
Point 10: Line 55-59 is too long, authors should fragment . sentence at productivity . shortages of ....
Response 10: Thanks for your comments. It is really true as your suggested. We are very sorry for our negligence. We have made correction according to your comments.
The changes to the manuscript are as follows:
“In Northeast China, high seasonal precipitation (70-80% occurring from June to September) and low average annual temperatures (4.0-5.5 °C) are salient climatic features, which are key factors limiting poplar growth and productivity. In addition, the lack of superior pop-lar varieties and clones is also the primary constraint restricting poplar cultivation in Northeast China”.
Please see line 123-127 in the revised manuscript.
Special thanks to you for your good comments.
Point 11: In the introduction, the authors fail to highlight the importance of poplar which makes validates the study. Why is poplar important to human or the ecology.
Response 11: Thanks for your comments. It is really true as your suggested. We are very sorry for our negligence. We have made correction according to your comments.
The changes to the manuscript are as follows:
“Genetic improvement research on poplar has been conducted for nearly 80 years, and many improved varieties have been selected and used widely, playing an important role in pulpwood, plywood, medium density fiberboard, water conservation effects, and road greening”.
Please see line 103-106 in the revised manuscript.
Special thanks to you for your good comments.
Point 12: Which of the studied parameters contribute to wood quality? please add this is the introduction.
Response 12: Thanks for your comments. It is really true as your suggested. We are very sorry for our negligence. We have made correction according to your comments.
The changes to the manuscript are as follows:
“Polyploid breeding has great application potential in the breeding of new forest varieties with fast growth, high quality and high stress resistance. Many research results have in-dicated that triploid poplar has a higher photosynthetic index and growth traits than diploid and tetraploid trees. Wang, et al found that there was a correlation between photosynthetic traits and wood property traits. Tang, et al found that growth traits was positively correlated with wood traits, and the indirect evaluation of polyploid materials through photosynthetic and growth traits could provide guarantee for the improvement of wood yield and quality”.
Please see line 112-119 in the revised manuscript.
Special thanks to you for your good comments.
Point 13: Line 67, include a full stop after......investigated . Then start several.......as a new sentence.
Response 13: Thanks for your comments. It is really true as your suggested. We are very sorry for our negligence. We have made correction according to your comments.
The changes to the manuscript are as follows:
“In this study, the growth traits and photosynthetic characteristics of 19 poplar clones of different ploidies were investigated. Several genetic and variation parameters were calculated, and elite clones were ultimately selected based on principal analysis”.
Please see line 135 in the revised manuscript.
Special thanks to you for your good comments.
Point 14: The authors need to review on the benefits conferred to trees through polyploidy.
Response 14: Thanks for your comments. It is really true as your suggested. We are very sorry for our negligence. We have made correction according to your comments.
Please see line 112-114 and line 121 in the revised manuscript.
Special thanks to you for your good comments.
Point 15: Line 76 the values of poplar growth are they with respect to years? or clones?
Response 15: Thanks for your comments. It is really true as your suggested. We are very sorry for our negligence. We have made correction according to your comments.
The changes to the manuscript are as follows:
“The contents of carbon, nitrogen, phosphorus, and potassium in soil were 16.62 mg/g, 1.65 mg/g, 0.66 mg/g and 18.63 mg/g, respectively”.
Please see line 144-145 in the revised manuscript.
Special thanks to you for your good comments.
Point 16: Line 86, what was the basis of selecting the 16 polyploidy poplar hybrid trees?
Response 16: Thanks for your comments. It is really true as your suggested. We are very sorry for our negligence. We have made correction according to your comments.
The changes to the manuscript are as follows:
“After three years of seedling growth, the hybrid progeny were preliminarily selected through seedling height and ground diameter, and 16 polyploid poplar hybrid clones with tree height and ground diameter over 20 % of the average were selected as super seedlings for further experiments”.
Please see line 267-269 in the revised manuscript.
Special thanks to you for your good comments.
Point 17: Line 93, check the correct name of the experimental design used by the authors.
In the experimental design,. did the authors factor in local control?
Response 17: Thanks for your comments. It is really true as your suggested. The experimental design name has been corrected. The soil conditions of the planted forestland were consistent, and the variance analysis was performed using the single plant data without considering the factor in local control.
Special thanks to you for your good comments.
Point 18: Description of the field experiment is vague. Did the authors evaluate the utilization potential of the 13 clones ? With the three diploid poplar clones as the controls? if so, why the six duplicate trees in the experiment?
Response 18: Thanks for your comments. It is really true as your suggested. We have made correction according to your comments.
13 triploid clones and 3 tetraploid clones were used as experimental materials, and 3 main cultivars were used as controls, and 30 trees from each clone and control were selected for the construction of experimental forest. For tree height, ground diameter and diameter height, all surviving healthy plants were investigated, but in the determination of leaf length, leaf width, leaf area, net photosynthesis rate, intercellular CO2 concentration, stomatal conductance, and transpiration rate, we selected 6 trees of per clone for measurement.
Special thanks to you for your good comments.
Point 19: The height and basal diameter and breast height were measured at what intervals?
Response 19: Thanks for your comments. It is really true as your suggested.
The investigation work was carried out after the end of the growing season and was performed every one year.
Special thanks to you for your good comments.
Point 20: The photosynthetic parameters were measured on sunny days: for how long?
Response 20: Thanks for your comments.
The photosynthetic parameters were measured on sunny days for two days.
Special thanks to you for your good comments.
Point 21: The variance analysis did not consider the blocking effect, was this an assumption that blocking had no effect ? if so, why didn't the authors use CRD instead?
Response 21: Thanks for your comments. It is really true as your suggested. The block effect was not considered in this study, and the experimental design of this study was completely random. The description of relevant content in the manuscript has been corrected.
Special thanks to you for your good comments.
Point 22: In their statistical model (line 121) the authors are not clear why they have effect due to clones (??) , at the same time, effect of the clone (??(?)). authors need to clarify.
Response 22: Thanks for your comments. It is really true as your suggested. We are very sorry for our negligence. We have made correction according to your comments.
The changes to the manuscript are as follows:
“where is the overall mean, is the effect of of ploidy type, is the effect value of the j clone in the i ploidy type, and is the random error”.
Please see line 324 in the revised manuscript.
Special thanks to you for your good comments.
Point 23: In line 163,, authors write that volume shared have the highest PCV or GCV values, reference should be provided.
Response 23: Thanks for your comments. It is really true as your suggested. We are very sorry for our negligence. We have made correction according to your comments.
We have added the reference in the manuscript. Please see line 373 in the revised manuscript.
Special thanks to you for your good comments.
Point 24: The footnote under table 2 should not stand for two tables. no matter the repetitiveness, every table and figure should stand alone.
Response 24: Thanks for your comments. It is really true as your suggested. We are very sorry for our negligence.
We have made correction according to your comments. We added the footnote to every table and figure.
Special thanks to you for your good comments.
Point 25: Table 4 , The authors divided the traits into three broad categories, instead of 'different traits' mention the three traits.
Response 25: Thanks for your comments. It is really true as your suggested. We are very sorry for our negligence. We have made correction according to your comments.
The changes are as follows:
“Table S3. Principal component analysis of three broad categories”.
Please see Table S3 in the revised manuscript.
Special thanks to you for your good comments.
Point 26: Table 5 title is incomplete . Tables should stand alone . authors should mention what the genetic gains are for
Response 26: Thanks for your comments. It is really true as your suggested. We have made correction according to your comments.
The changes are as follows:
“Table 4. Comprehensive evaluation of elite clones”.
Please see Table 4 in the revised manuscript.
Special thanks to you for your good comments.
Point 27: Discussion: 4.2 mean values , the authors results on Pn and WUE contradicted those on populus ussuriensis. Authors should briefly state the p. urruriensis findings.
Response 27: Thanks for your comments. It is really true as your suggested. We are very sorry for our negligence. We have made correction according to your comments.
The changes are as follows:
“However, the Pn and Wue values of the triploid were higher than those of the diploid and tetraploid; these results were the same as those of research on hybrid poplar clones, but in Populus ussuriensis, triploid is the largest, diploid is the smallest, and tetraploid is the center.
Please see line 570-573 in the revised manuscript”.
Special thanks to you for your good comments.
Point 28: Line 370 sounds incomplete, consider revising it
Response 28: Thanks for your comments. It is really true as your suggested. We are very sorry for our negligence. We have made correction according to your comments.
The changes to the manuscript are as follows:
“In our study, different principal components representative different categories of investi-gated traits, such as PCI represented growth traits, PCII represented leaf traits, and PCIII represented photosynthetic traits”.
Please see line 799-801 in the revised manuscript.
Special thanks to you for your good comments.
Point 29: Line 386-389, interpret the study findings with the information in line 384-386
Response 29: Thanks for your comments. It is really true as your suggested. We are very sorry for our negligence. We have made correction according to your comments.
The changes to the manuscript are as follows:
“In this study, compared with the other clones, three excellent clones had higher genetic gains in growth indicators, leaf indicators and photosynthetic indicators. This indicated that the elite clones had good growth performance which could provide superior material for timber production”.
Please see line 817-820 in the revised manuscript.
Special thanks to you for your good comments.
Point 30: More comments and suggestions are in the manuscript attached.
Response 30: Thanks for your comments. It is really true as your suggested. We are very sorry for our negligence. We have made correction according to your comments.
Special thanks to you for your good comments.
Reviewer 3 Report
The presented research is interesting, though the presentation of the results needs to be improved. In general, the results section is too technical and hard to read, and much of the material that should be in the introduction is given in the discussion section. I think fixing these issues would improve the readability of the manuscript significantly. Here are some minor technical issues that need to be addressed.
1. in the abstract, uncommon abbreviations should not be used without defining them, ie. Pn in line 19, phenotypic traits in lines 30 and 31.
2. P1L25 "were significantly positively correlated with each other but negatively positively correlated" which one is it - negative or positive
3. P2L75, the temperature is given in ts? What is ts?
4. P2L76-77, "These factors led to slow poplar growth of..." it is not clear what are those numbers and how they apply to the presented work. Is it the growth of those already chosen genotypes from the 80's or different locations in this area?
5. Table 1 is unreadable since the vertical alignment in the first column is in the middle, so I cannot say which of the samples were the control in this study
6. P3.L103 says "leaf weight," but it is in mm; is it maybe width?
7. P3.L104 please explain how is the area of the leaf measured with the digital caliper?
8. P3.L116, "using" was written in superscript
9. P4.L160 Significance is usually given as p value
10. P8, it would be more informative to give PCA scatter plot than table 4. Table 4 could be given as supplemental material. Regarding this analysis, what kind of data was used for this analysis - raw data or transformed? Considering the different nature of the parameters used, I would presume the data was transformed, but I cannot find any information in the manuscript.
11. P12L370 "Different principal components instead of different investigated traits. " What does this sentence means? It looks like a part of a sentence.
Author Response
Response to Reviewer 3 Comments
Thank you for your comments. These comments are all valuable and very helpful for revising and improving our paper, as well as the important guiding significance to our researches. We have studied comments carefully and have made correction which we hope meet with approval. The reply is as follows.
Point 1: in the abstract, uncommon abbreviations should not be used without defining them, ie. Pn in line 19, phenotypic traits in lines 30 and 31.
Response 1: Thanks for your comments. It is really true as your suggested. We are very sorry for our negligence. We have made correction according to your comments.
Please see line 21, and line 35-38 in the revised manuscript.
Special thanks to you for your good comments.
Point 2: P1L25 "were significantly positively correlated with each other but negatively positively correlated" which one is it - negative or positive
Response 2: Thanks for your comments. It is really true as your suggested. We are very sorry for our negligence. We have made correction according to your comments.
The changes to the manuscript are as follows:
“Transpiration rate, intercellular carbon dioxide concentration and stomatal conductance were significantly positively correlated with each other, but negatively correlated with instantaneous water use efficiency.”
Please see line 29 in the revised manuscript.
Special thanks to you for your good comments.
Point 3: P2L75, the temperature is given in ts? What is ts?
Response 3: Thanks for your comments. It is really true as your suggested. We are very sorry for our spelling mistake. We have made correction according to your comments.
The changes to the manuscript are as follows:
“The precipitation ranged from 420 to 480 mm, and the annual average temperature was 3.4 ℃ in poplar and the ecological environment in arid areas”.
Please see line 143 in the revised manuscript.
Special thanks to you for your good comments.
Point 4: P2L76-77, "These factors led to slow poplar growth of..." it is not clear what are those numbers and how they apply to the presented work. Is it the growth of those already chosen genotypes from the 80's or different locations in this area?
Response 4: Thanks for your comments. It is really true as your suggested. We are very sorry for our negligence. We have made correction according to your comments.
The changes to the manuscript are as follows:
“The contents of carbon, nitrogen, phosphorus, and potassium in soil were 16.62 mg/g, 1.65 mg/g, 0.66 mg/g and 18.63 mg/g, respectively”.
Please see line 144-145 in the revised manuscript.
Special thanks to you for your good comments.
Point 5: Table 1 is unreadable since the vertical alignment in the first column is in the middle, so I cannot say which of the samples were the control in this study
Response 5: Thanks for your comments. It is really true as your suggested. We are very sorry for our negligence. We have made correction according to your comments.
Please see Table 1 in the revised manuscript.
Special thanks to you for your good comments.
Point 6: P3.L103 says "leaf weight," but it is in mm; is it maybe width?
Response 6: Thanks for your comments. It is really true as your suggested. We are very sorry for our negligence. We have made correction according to your comments.
Please see line 286 in the revised manuscript.
Special thanks to you for your good comments.
Point 7: P3.L104 please explain how is the area of the leaf measured with the digital caliper?
Response 7: Thanks for your comments. It is really true as your suggested. We are very sorry for our negligence. We have made correction according to your comments.
The changes to the manuscript are as follows:
“Leaf length and leaf width were measured by vernier caliper, and leaf area was measured by ImageJ software”.
Please see line 286-287 in the revised manuscript.
Special thanks to you for your good comments.
Point 8: P3.L116, "using" was written in superscript
Response 8: Thanks for your comments. It is really true as your suggested. We are very sorry for our negligence. We have made correction according to your comments.
Please see line 318 in the revised manuscript.
Special thanks to you for your good comments.
Point 9: P4.L160 Significance is usually given as p value
Response 9: Thanks for your comments. It is really true as your suggested. We are very sorry for our negligence. We have made correction according to your comments.
Please see line 370 in the revised manuscript.
Special thanks to you for your good comments.
Point 10: P8, it would be more informative to give PCA scatter plot than table 4. Table 4 could be given as supplemental material. Regarding this analysis, what kind of data was used for this analysis - raw data or transformed? Considering the different nature of the parameters used, I would presume the data was transformed, but I cannot find any information in the manuscript.
Response 10: Thanks for your comments. It is really true as your suggested. We are very sorry for our negligence. We have made correction according to your comments.
The changes to the manuscript are as follows:
“Principal component analysis (PCA) was performed after data transformation, three prin-cipal components with high eigenvalues were obtained with a cumulative contribution rate of 81.30% (Table S3)”.
Please see line 479 in the revised manuscript.
Special thanks to you for your good comments.
Point 11: P12L370 "Different principal components instead of different investigated traits. " What does this sentence means? It looks like a part of a sentence.
Response 11: Thanks for your comments. It is really true as your suggested. We are very sorry for our negligence. We have made correction according to your comments.
The changes to the manuscript are as follows:
“In our study, different principal components representative different categories of investi-gated traits, such as PCI represented growth traits, PCII represented leaf traits, and PCIII represented photosynthetic traits”.
Please see line 799-801 in the revised manuscript.
Special thanks to you for your good comments.